Dental pulp stem cells ameliorate D-galactose-induced cardiac ageing in rats

El-Akabawy Gehan gehanakabawy@gmail.com 1 2 3
El-Kersh Sherif Othman Fathy 4
El-Kersh Ahmed Othman Fathy Othman 5
Amin Shaimaa Nasr 6 7
Rashed Laila Ahmed 8
Abdel Latif Noha 9 10
Elshamey Ahmed 11
Abdallah Mohamed Abdallah Abd El Megied 12
Saleh Ibrahim G. 13 14
Hein Zaw Myo 1 2
El-Serafi Ibrahim 1
Eid Nabil 15
1 Department of Basic Medical Sciences, College of Medicine, Ajman University , Ajman , United Arab Emirates
2 Centre of Medical and Bio-allied Health Sciences Research, Ajman University , Ajman , United Arab Emirates
3 Department of Anatomy and Embryology, Faculty of Medicine, Menoufia University , Menoufia , Egypt
4 Faculty of Medicine, Galala University , Suez , Egypt
5 Department of Periodontology, Faculty of Dentistry, Kafr El Sheikh University , Kafr El Sheikh , Egypt
6 Department of Anatomy, Physiology and Biochemistry, Faculty of Medicine, The Hashemite University , Zarqa , Jordan
7 Department of Physiology, Faculty of Medicine, Cairo University , Cairo , Egypt
8 Department of Medical Biochemistry, Faculty of Medicine, Cairo University , Cairo , Egypt
9 Department of Medical Pharmacology, Faculty of Medicine, Cairo University , Cairo , Egypt
10 Department of Medical Pharmacology, Armed Forces College of Medicine , Cairo , Egypt
11 Samanoud General Hospital, Samannoud City, Samanoud , Gharbia , Egypt
12 Department of Paediatrics, Cairo University, Faculty of Medicine , Cairo , Egypt
13 Department of Pharmacology and Toxicology, Faculty of Pharmacy, Al-Azhar University , Cairo , Egypt
14 Department of Clinical Pharmacy and Pharmacy Practice, Faculty of Pharmacy, Sinai University, Kantra , Ismailia , Egypt
15 Department of Anatomy, Division of Human Biology, School of Medicine, International Medical University , Kuala Lumpur , Malaysia
Fernandez Anne
Electronic publication date: 2024 May 21
Publication date: 2024
Volume: 12
Electronic Location ID: e17299
Received 2023 Nov 30; Accepted 2024 Apr 3
Copyright: ©2024 El-Akabawy et al.
Copyright year: 2024
Copyright holder: El-Akabawy et al.
License: This is an open access article distributed under the terms of the Creative Commons Attribution License, which permits unrestricted use, distribution, reproduction and adaptation in any medium and for any purpose provided that it is properly attributed. For attribution, the original author(s), title, publication source (PeerJ) and either DOI or URL of the article must be cited.
License URL: https://creativecommons.org/licenses/by/4.0/

Keywords: Dental pulp stem cell, Cardiac aging, Rat model, D-galactose

Funding: Deanship of Research and Graduate Studies, Ajman University, UAE 2022-IRG-MED-5 This research project was funded by the Deanship of Research and Graduate Studies, Ajman University, UAE (grant no. 2022-IRG-MED-5). The funders had no role in study design, data collection and analysis, decision to publish, or preparation of the manuscript.

==============================
Background

Ageing is a key risk factor for cardiovascular disease and is linked to several alterations in cardiac structure and function, including left ventricular hypertrophy and increased cardiomyocyte volume, as well as a decline in the number of cardiomyocytes and ventricular dysfunction, emphasizing the pathological impacts of cardiomyocyte ageing. Dental pulp stem cells (DPSCs) are promising as a cellular therapeutic source due to their minimally invasive surgical approach and remarkable proliferative ability.

Aim

This study is the first to investigate the outcomes of the systemic transplantation of DPSCs in a D-galactose (D-gal)-induced rat model of cardiac ageing. Methods. Thirty 9-week-old Sprague-Dawley male rats were randomly assigned into three groups: control, ageing (D-gal), and transplanted groups (D-gal + DPSCs). D-gal (300 mg/kg/day) was administered intraperitoneally daily for 8 weeks. The rats in the transplantation group were intravenously injected with DPSCs at a dose of 1 × 106 once every 2 weeks.

Results

The transplanted cells migrated to the heart, differentiated into cardiomyocytes, improved cardiac function, upregulated Sirt1 expression, exerted antioxidative effects, modulated connexin-43 expression, attenuated cardiac histopathological alterations, and had anti-senescent and anti-apoptotic effects.

Conclusion

Our results reveal the beneficial effects of DPSC transplantation in a cardiac ageing rat model, suggesting their potential as a viable cell therapy for ageing hearts.

Introduction

Ageing is a progressive, multifaceted process that is linked to decreased physiological performance and an elevated risk of mortality (Guo et al., 2022). Although cardiac ageing is a natural process, it is linked to various pathological factors that may result in age-related disorders (Redgrave et al., 2023). Extensive research has demonstrated that ageing is linked to changes in cardiac morphology and function, such as an increase in left ventricular hypertrophy and cardiomyocyte volume as well as a decrease in cardiomyocyte number and ventricular dysfunction, emphasizing the pathological impacts of cardiomyocyte senescence (Dai et al., 2012; Shimizu & Minamino, 2019).

A substantial body of evidence indicates that cardiovascular diseases and mechanisms involved in cellular senescence are closely related (Pagan et al., 2022). The ageing heart is accompanied by mitochondrial malfunction, and the consequent generation of reactive oxygen species (ROS) may lead to age-related cardiac dysfunction (Shimizu & Minamino, 2019). This generation of ROS in cardiac tissue triggers the upregulated expression of ageing markers p53, p21, and p16 (Shimizu & Minamino, 2019). These markers are cellular regulatory factors involved in the cell cycle and are remarkably upregulated in senescent cells. P53, a key player in the DNA damage response (DDR) pathway, can induce either temporary or permanent cellcycle arrest (cellular senescence). When presented with stimuli triggering DDR, cell-growth arrest and senescence are mediated via the murine double minute 2 (MDM2)-P53-P21 and P16-retinoblastoma protein (pRb) pathways. Administration of D-galactose results in the elevated expression of P53, P21, and P16 (Bei et al., 2018; Wang et al., 2022). Sirt1, which is a member of a group of NAD-dependent deacetylases collectively referred to as sirtuins, is involved in various molecular pathways and is recognized as a pivotal protein in the control of ageing and metabolism. Several reports have demonstrated that Sirt1 has a role in regulating cardiac myocyte growth and survival under stress (Alcendor et al., 2007; Cencioni et al., 2015; Lu et al., 2014; Ministrini et al., 2021). Furthermore, an imbalance in apoptotic control may be a major factor in ageing. Mitochondria generate apoptogenic substances, such as cytochrome c, which regulate cellular apoptosis. The ratio of pro- and anti-apoptotic Bcl-2 family members, or pro- and anti-apoptotic proteins, is critical in the intrinsic apoptosis process (Phaneuf & Leeuwenburgh, 2002; No et al., 2020).

Mesenchymal stem cells (MSCs) have exhibited encouraging therapeutic prospects for numerous disorders and organ repairs. Adult stem cells have been successfully harvested from various organs and can differentiate into various cell phenotypes. Their implantation presents a limited risk of tumorigenesis and poses almost no ethical issues (Brown et al., 2019). MSCs have been harvested from bone, adipose, lung, umbilical cord, and dental pulp tissue (Berebichez-Fridman & Montero-Olvera, 2018).  Growing experimental and clinical evidence supports that MSCs constitute a promising therapeutic approach for the remedy of cardiac dysfunction. They mediate the production of a plethora of growth factors, replace lost cells, and create a niche that promotes endogenous cardiac repair (Bahrami et al., 2023; Barrere-Lemaire et al., 2023; Bui et al., 2023; Gu et al., 2023). Limited studies have investigated the potential beneficial effects of MSC injection in animal models of ageing hearts. In rodent ageing models, bone marrow- and adipose-derived MSCs have proven their efficacy in improving cardiac function and downregulating ageing-linked cardiac damage, which is associated with a downregulation in the expression of apoptosis and senescence markers. However, in these studies, homing, survival, and potential differentiation of MSCs into cardiomyocytes have not been examined (Zhang et al., 2015; Chang et al., 2021).

Considering the long-term clinical benefits, autologous MSC transplantation is superior to allogeneic cell transplantation. However, the outcomes of autologous cell therapy remarkably deteriorate with age due to their depletion and limited self-renewal capacity. Recent research has highlighted the impact of ageing and illnesses on both human and rodent tissues, demonstrating phenotypic and functional alterations in endogenous MSCs derived from the bone marrow, adipose tissue, and heart (Li et al., 2013; Gnani et al., 2019; Hong et al., 2020; Guo et al., 2023). Conversely, recent studies have shown that dental pulp stem cells (DPSCs) exhibit resistance to senescence effects and possess enhanced differentiation potential in vitro and regeneration capabilities in vivo compared to bone marrow mesenchymal stem cells (BMMSCs) (Ma et al., 2019). Furthermore, DPSCs isolated from elderly individuals demonstrate active metabolism; their derived miRNAs and exosomes represent a rich source of nanovesicles for the treatment of age-related disorders, indicating the healthy condition of these cells and emphasizing their suitability for autologous applications (Jezzi et al., 2019). Such suitability, which is attributed to the resistance of these cells to ageing process, indicates that they are a promising option for the treatment of age-related diseases.

A previous study provided the first proof-of-principle that the DPSC secretome protects against D-gal-induced ageing of multiple organs (Kumar et al., 2022). The D-galactose-induced ageing model is well-established and widely recognized. D-galactose treatment can efficiently induce cardiac ageing as indicated by elevated levels of several cardiac ageing markers, including oxidative stress, decreased expression of antioxidants such as superoxide dismutase (SOD), upregulated levels of p53, and enhanced cardiac apoptosis (Bo-Htay et al., 2018; Wang et al., 2022). However, the efficacy of the systemic administration of DPSCs in ameliorating age-associated cardiac function and structural deterioration has not yet been evaluated. Several studies have reported the capability of DPSCs to differentiate into cardiomyocytes in vitro (Xin et al., 2013; Sung et al., 2016); however, to the best of our knowledge, such trans-differentiation abilities have not yet been evaluated in ageing hearts. In the current study, we aimed to assess, for the first time, the possible efficacy of an intravenous injection of DPSCs in a D-gal-induced rat model of cardiac ageing to evaluate their potential as a preventive therapy for age-associated cardiovascular diseases.

Materials & Methods

Animals

Thirty male Sprague–Dawley rats (8 weeks old, 180–200 g) were purchased from the Theodor Bilharz Research Institute, Imbaba, Egypt, and kept in the animal house of the Faculty of Medicine, Menoufia University, Egypt as previously described by (El-Akabawy et al., 2023). The animals were acclimatized to laboratory conditions for 1 week before the start of the experiment. The rats were kept in standard cages: two were rats kept in each cage, under standard laboratory conditions (22 ± 5 °C, 60 ± 5% humidity, and a 12-h/12-h light/dark cycle). Standard laboratory chow and tap water were provided ad libitum. At the end of the experiment, rats were anaesthetized via an intraperitoneal injection of ketamine (90 mg/kg) and xylazine (15 mg/kg) and were decapitated (El-Akabawy et al., 2023). Rats were euthanized in the event of rapid weight loss or impaired ambulation  via lethal injection of pentobarbital sodium (200 mg/kg). In this study, no rats were euthanized prior to the planned end of the experiment. All experimental procedures involving animals were approved by the Institutional Review Board of Ajman University, UAE [IRB# M-F-A-11-Oct].

DPSC isolation and culture

Dental pulp tissues were obtained from the pulpal cavity of Sprague–Dawley male rat incisors (aged 6–8-week-old) and cultured as previously described (Patel et al., 2009). Briefly, the dental pulp tissues were promptly harvested and enzymatically digested in a solution of 3 mg/mL collagenase type 1 (Sigma-Aldrich, St. Louis, MO, USA) for 1 h at 37 °C. The cells (1 × 106 cells) were cultured in T25 cm flasks (Falcon). The culture media consisted of Dulbecco’s Modified Eagle Medium (DMEM) supplemented with 20% foetal bovine serum (FBS, Gibco), 100 U/mL penicillin, and 100 µg/mL streptomycin. The flasks were incubated at 37 °C in a humidified incubator with 5% CO2. Media were replaced every three days. The cells were passaged at 80% confluence using 0.05% trypsin–EDTA (Sigma-Aldrich) for 3–5 min. To evaluate cell viability, the cell suspension was mixed with 0.4% Trypan blue (Gibco), and 10 µL of the mixture was loaded in each chamber of a haemocytometer. Counting of the viable and non-viable cells was conducted within 5 min. Cells of passage 4 were evaluated.

Flow cytometry

Cells were resuspended in staining buffer (2% FBS/ phosphate buffered saline (PBS)) and surface-stained with FITC-conjugated mouse anti-rat CD105 (BioLegend, UK), FITCH-conjugated mouse anti-rat CD90 (BD Pharmingen, San Diego, CA, USA), PE-conjugated rabbit anti-rat CD34 (Abcam, Cambridge, UK), or PE-conjugated rabbit anti-rat CD45 (Abcam, Cambridge, UK) at 4 °C for 30 min. Isotype-matched antibodies were used as controls. Cells were analysed using an EPICS XL flow cytometer (Beckman Coulter, Brea, CA, USA) (El-Akabawy et al., 2022).

Experimental design

Rats were randomly divided into three groups: control, D-galactose (D-gal)-treated, and D-gal + DPSCs-treated ( n = 10 in each group). G Power software was used to determine the sample size. Rats (aged 9 weeks) in the D-gal- and D-gal + DPSCs-treated groups were given intraperitoneal injections of d-gal (300 mg/kg, Sigma-Aldrich, St. Louis, MO, USA) daily for 8 weeks. Rats in the D-gal + DPSCs group received intravenous administration into the tail vein of 1 × 106 DPSCs labelled with the membrane-bound fluorescent marker PKH26 (Sigma-Aldrich, St. Louis, MO, USA) once every two weeks (El-Akabawy et al., 2022).

Measurement of body weight and the heart index

Body weights were measured weekly. At the end of the experiment, the rats were anaesthetized via intraperitoneal injection of ketamine (90 mg/ /kg) and xylazine (15 mg/kg) and were decapitated. Hearts were immediately dissected and weighed. Heart indices were calculated as follows: heart tissue weight (mg)/final body weight (g) (El-Akabawy et al., 2022).

Transthoracic echocardiography

All transthoracic echocardiography (TTE) measurements were performed using a linear transducer. A linear-array probe was used at a frequency of 10 MHz and attached to a Mindray M7 premium (Shenzhen Mindray Bio-Medical Electronics Co., Ltd., PR China) ultrasound echocardiography Doppler machine. The rats were anaesthetized by intraperitoneal injection of ketamine hydrochloride (25 mg/kg) and xylazine (5 mg/kg). Anaesthesia was followed by hair removal from the anterior part of the chest and thereafter, rats were kept on a specialized warming table to sustain normothermia (Watson et al., 2019).

Ejection fraction (EF) and fractional shortening (FS), left ventricular internal dimension at end-systole (LVIDs), left ventricular end-systolic volume (ESV), left ventricular posterior wall thickness at end-diastole (LVPWd), left ventricular internal dimension at end-diastole (LVIDd), and left ventricular end-diastolic volume (EDV) were assessed following the American Society of Echocardiography guidelines. This assessment was performed in a blind manner by an independent experienced researcher.

Electrocardiogram recordings

Three touch electrodes of the PowerLab settings (MLA1214, AD Instruments, New South Wales, Australia) were attached to an animal bioamplifier (FE136 Animal Bio Amp, AD instruments, New South Wales, Australia) and fixed to the animals. Lead placement was conducted according to the lead II configuration for determining heart rate (HR) in small laboratory animals (Stohr, 1998).

Electrocardiogram (ECG) recordings were performed as previously described by Harkin, O’Donnell & Kelly (2002). A stretch of the ECG readings was obtained to calibrate the second channel for simultaneous beats per minute (BPM) recordings. The ‘Ratemeter’ (within Chart for Powerlab; Holliston, MA, USA) was utilized simultaneously to measure the HR on channel 2 from the ECG plot in channel 1. HR was adjusted between 0 and 500 BPM. The ratemeter band was adjusted in such a way that the upper boundary remained positioned lower than the R-wave peak, whereas the lower line was higher than the P- and T-waves and any other noise. This facilitated the monitoring of the ECG waves on channel 1 with simultaneous BPM recorded on channel 2. The ECG was recorded at a sampling speed of 500/s and within a voltage range of 500 mV. A high-pass filter was adjusted to 0.3 Hz, and a low-pass filter (50 Hz) was used.

The ECG were digitized and stored using standard PC-based hardware (AD Instruments, Dunedin, New Zealand). PowerLab v.7.3.7 was utilized to illustrate the recording diagrams. The recordings were analysed using LabChart software (AD Instruments, Dunedin, New Zealand). The recorded ECG was used to calculate R-R, QRS, PR, QT, cQT, and ST intervals using the same software. Throughout the experiment, the rats’ body temperature was sustained at 38 °C. The time at the start of the recording was set as 0.0 min. The corrected QT (cQT) was determined using Bazett’s formula [QTc = QT/RR1/2] (Goldenberg, Moss & Zareba, 2006) installed in the software.

Assessment of oxidative stress and antioxidant indices

Malondialdehyde (MDA) and glutathione (GSH) levels, as well as superoxide dismutase (SOD) activity, were measured in the cardiac tissue using a spectrophotometer. Rat cardiac tissues (100 mg) were homogenized in one mL of phosphate-buffered saline (PBS; pH 7.0) to assess the MDA level and determine the degree of lipid peroxidation. After mixing the homogenates with 20% trichloroacetic acid (TCA), the mixtures were centrifuged at 5000 rpm for 15 min. A 5% thiobarbituric acid (TBA) solution was added to the supernatants and boiled for 10 min. The absorbance was obtained at 532 nm and a standard curve was used to quantify the MDA levels. The results are presented as nanomoles (nmol) per milligram (mg) of protein.

Based on the suppression of a nitro blue tetrazolium reduction by O2 produced by the xanthine/xanthine oxidase system, the superoxide dismutase (SOD) activity was determined, and absorbance was obtained at 550 nm. The findings are represented as units (U) per milligram (mg) of protein. One SOD activity unit was considered as the enzyme concentration required to generate 50% inhibition in one mL reaction solution per mg of tissue protein.

To examine GSH concentrations, cardiac tissue homogenates were incubated with a solution of dithiobis nitrobenzoate (DTNB) for 1 h. The absorbance was obtained at 412 nm. A standard curve was used to determine the GSH level. The results are presented in micromoles (mmol) per mg of protein.

Quantitative reverse-transcription polymerase chain reaction

RNA was isolated from cardiac-tissue homogenates of rats in each group using the RNeasy Purification Reagent (Qiagen, Hilden, Germany). RNA purity was assessed using a spectrophotometer, ensuring a 260/280 nm absorption ratio of 1.8–2.0 for all samples. Subsequently, cDNA synthesis was performed employing Superscript II (Gibco Life Technologies, Waltham, MA, USA). Quantitative PCR (qPCR) was performed on a StepOneTM instrument with software version 3.1 (Applied Biosystems, Foster City, CA, USA). Reaction mixtures contained SYBR Green Master Mix (Applied Biosystems, Foster City, CA, USA), gene-specific primer pairs (detailed in Table 1), cDNA, and nuclease-free water. Cycling conditions comprised an initial denaturation step at 95 °C for 10 min, followed by 40 cycles of 15 s at 95 °C and 60 s at 60 °C. Data analysis was conducted using the ABI Prism sequence detection system software, and quantification was performed with the Sequence Detection Software v1.7 (PE Biosystems, Foster City, CA, USA). The comparative cycle threshold method (Livak & Schmittgen, 2001) was used to determine relative expression levels of the target gene, with all values normalized to β-actin mRNA.

Table 1 List of primers used in RT-qPCR.

Gene	Accession number	Forward and reverse primer sequences (5′ → 3′)	
P53	NM_030989.3	CCAGGATGTTGCAGAGTTGTTAGA	
		TTGAGAAGGGACGGAAGATGAC	
P21	U24174.1	GGGACAGCAGAGGAAGACC	
		GACTAAGGCAGAAGATGTAGAGC	
P16	L81167.1	CTCCTTGGCTTCATTCTGG	
		TCCAATCGTCTCCCTCCCTC	
BETA ACTIN	NM_031144	ATTTGGCACCACACTTTCTACA	
		TCACGCACGATTTCCCTCTCAG	

Western blot analysis

Western blot analysis was conducted as previously described before by (Al-Serwi, El-Kersh & El-Akabawy, 2021). Using radioimmunoprecipitation buffer (Sigma-Aldrich, St. Louis, MO, USA), proteins were isolated from the cardiac tissues. The homogenates were centrifuged at 12,000 × g at 4 °C for 20 min and the protein level was measured in lysate aliquots using a protein assay kit (Bio-Rad, Hercules, CA, USA). Samples were boiled at 95 °C for 5 min, separated (20 µg/lane) using 7% sodium dodecyl sulphate–polyacrylamide gel electrophoresis, and subsequently transferred to nitrocellulose membranes (Bio-Rad, Hercules, CA, USA). Next, the membranes were blocked for 1 h at room temperature (RT) using 5% bovine serum albumin in Tris-buffered saline (TBS), and then incubated for 12 h at 4  °C with primary antibodies specific for anti-sirt1 (Abcam, Cambridge, UK, cat # ab110304), anti-cleaved caspase-3 (Abcam, Cambridge, UK, cat # ab184787), anti-cytochrome c (Abcam, Cambridge, UK, cat # ab133504), anti-Bax (Abcam, Cambridge, UK, cat # ab32503), and anti-Bcl-2 (Abcam, Cambridge, UK, cat ab194583). After rinsing with TBS, the membranes were incubated with secondary horseradish peroxidase-conjugated anti-rabbit IgG or anti-mouse IgG antibody (1: 3000, Bio-Rad) for 1 h at RT. Proteins were visualized using enhanced chemiluminescence (ECL Plus; Amersham, Arlington Heights, IL, USA) and measured via densitometry using Molecular Analyst Software (Bio-Rad). The relative expression of each protein band was normalized to that of β-actin.

Histological and immunohistochemical analyses

At the end of the experiment, cardiac tissues were dissected and fixed in 10% formalin and embedded in paraffin wax. For histological assessment, 5-µm left ventricular sections were de-paraffinized, rehydrated using a graded ethanol series (100%, 90%, and 70%), and stained with haematoxylin and eosin (H&E) or Masson’s trichrome.

For immunofluorescence staining, cardiac tissue was fixed at 4  °C for 24 h and cryoprotected in 30% sucrose at 4  °C. Using a cryostat, serial sections (40 µm) were obtained and kept at −20 °C until use. The sections were incubated in 10% blocking solution at RT for 30 min and then incubated at 4 °C overnight with the following primary antibodies; rabbit Anti-Cardiac Troponin T antibody (1:2000, Abcam, Cat. #ab209813), rabbit Anti-Cardiac Troponin I antibody (1:1000, Abcam, Cat. #ab209809), or rabbit anti-connexin-34 (1:1000, Abcam, Cat. #ab259276). Thereafter, sections were rinsed with PBS and incubated with a secondary antibody (1:500, Alexa-488, Cat. #A-11034, Molecular Probes) for 1 h. After washing with PBS, the sections were mounted in Fluoroshield mounting medium containing DAPI (Abcam, Cat. #ab104139).

Quantitative histological assessments

For quantitative evaluation, three H&E- and Masson-stained sections per rat were used. By applying ImageJ software (NIH, Bethesda, MD, USA), the H&E- and Masson-stained sections were examined to determine the cardiomyocyte area and Masson’s-stained area, respectively. A Leica DMLB2/11888111 microscope equipped with a Leica DFC450 camera was used to acquire the images.

Connexin-43 immunofluorescence was measured by randomly capturing five non-overlapping images per slide. A Leica DM5500 B/11888817/12 microscope, fitted with a Leica HI PLAN 10×/0.25 objective and a Leica DFC450C camera, was used to capture the images. Connexin-43-stained spots were manually counted using the plugin/cell counting tool (Rangan & Tesch, 2007) in ImageJ software (National Institutes of Health, Bethesda, MD, USA), and the average per field for each rat was then calculated. This measurement was performed in a blind manner by an independent experienced researcher. For statistical analysis and comparison, ten animals were used per experimental group.

Statistical analysis

Statistical analysis was performed as previously described by El-Akabawy et al. (2022). The data are presented as the mean ± SEM. Normal distributions were assessed using the D’Argostino and Pearson normality tests, and data were analysed using one- or two-way analysis of variance (ANOVA), followed by a post hoc Bonferroni test. P <0.05 was considered statistically significant. Statistical analyses were performed using GraphPad Prism 5.03 software (GraphPad Software, San Diego, CA, USA). No inclusion/exclusion criteria were applied in this study.

Results

Characterization of DPSCs

DPSCs derived from the dental pulp tissue of Sprague–Dawley rats were spindle-shaped after 10 days of culture. Flow cytometry was used to characterize the cells at passage 4. The expression of CD90 and CD105 (mesenchymal cell markers), and CD45 and CD34 (hematopoietic lineage markers) were assessed. Over 90% of the cells were identified as CD90+ and CD105+ and less than 10% were identified as CD45+ and CD34+ (Fig. S1). These findings suggest that most of these cells were MSCs.

DPSC transplantation improves body weight and heart indices

Body weights of the rats in the control, D-gal and transplanted groups did not differ significantly (Fig. 1A). However, the heart index was dramatically increased in aged rats than in control rats, suggesting cardiac hypertrophy. DPSC transplantation dramatically decreased the heart index compared with that of D-gal rats (Fig. 1B), indicating that the transplanted cells attenuated D-gal-induced hypertrophy.

Figure 1 Body weight (A) and heart index (B) were assessed in control, aged (D-gal), and transplanted (D-gal + DPSCs) rats.

**P < 0.001 vs. control rats; #P < 0.01 vs. aged rats. Data are expressed as means ± SEM. N = 10 per group.

Intravenous injection of DPSCs reduces cardiac dysfunction in D-galactose-induced aged rats

Next, we evaluated the effects of DPSC transplantation on D-galactose (D-gal)-induced cardiac ageing in rats using echocardiography. The results revealed widespread LV systolic and diastolic dysfunction, including reduced EF% and FS%, elevated LVIDs, ESV, and LVPWd in D-gal-treated rats. Compared to D-gal-treated rats, DPSC injection improved EF% (Fig. 2B) and FS% (Fig. 2C) and reduced LVPWd (Fig. 2D), LVIDs (Fig. 2E), ESV (Fig. 2F) measurements. We observed an increase in left ventricular internal dimension at end-diastole (LVIDd) (Fig. 2G) and left ventricular end-diastolic volume (EDV) (Fig. 2H) in aged hearts; however, this difference was not statistically significant. D-gal intraperitoneal injection led to significant changes in the ECG results of treated rats in the form of increased QRS duration and cQT interval, but decreased R amplitude and ST height compared to the control group. Rats in the D-gal + DPSCs group exhibited an improved ECG pattern, as indicated by a significantly decreased QRS duration and cQT interval, and an elevated R amplitude and ST height, compared to the aged group (Table 2).

Figure 2 DPSCs attenuate cardiac function alterations in D-gal-induced cardiac ageing at 2 weeks post-transplantation after the last DPSC injection.

Representative echocardiographic images from rats of control, D-gal, and D-gal + DPSCs groups (A). Ejection fraction (EF)% (B), fraction shortening (FS)% (C), left ventricular posterior wall thickness at end-diastole (LVPWd) (D), left ventricular internal dimension at end systole (LVIDs) (E), left ventricular end-systolic volume (ESV) (F), left ventricular internal dimension at end-diastole (LVIDd) (G), and left ventricular end-diastolic volume (EDV) (H) were measured using echocardiography. *P < 0.05, **P < 0.01, ***P < 0.001, compared with the control group; #P < 0.05, ##P < 0.01, compared with the D-gal group. Data are expressed as mean ± SEM. N = 10 per group.

Effect of DPSC transplantation on the expression of connexin-43

The decreased production of the critical gap junction protein, connexin-43, plays a pivotal role in age-related cardiac dysfunction (Rodriguez-Sinovas et al., 2021). To determine whether DPSC transplantation restored connexin-43 expression in the hearts of aged rats, connexin-43 immunoreactive areas were evaluated. In D-gal-treated rats, immunoreactive connexin-43 areas were sparse, and their expression in cardiac tissue markedly decreased, whereas in the D-gal + DPSCs group, they were markedly upregulated (Figs. 3A–3D).

DPSC systemic transplantation attenuates cardiac histopathological alterations in D-galactose-induced aged rats

To examine the alterations in the myocardial architecture of different groups, left ventricular cardiac tissue slides were stained with H&E. The left ventricular cardiac tissue of D-gal-induced ageing rats exhibited a distorted myocardial structure, characterized by a disorganized arrangement of cardiomyocytes and expanded intercellular space (arrows, Fig. 4A) when compared to the control rats. In contrast, marked improvement was observed in the ageing rats that received DSPSC injection. H&E staining also revealed that the left ventricular cardiomyocyte cross-sectional area was significantly enlarged in D-gal-induced ageing rats, whereas DPSC treatment markedly decreased the cardiomyocyte area in D-gal + DPSC-treated rats (Fig. 4B). Masson’s trichrome staining was used to examine the degree of cardiac fibrosis in the different groups. The collagen-stained area in the interstitial and perivascular areas of the myocardium dramatically increased in D-gal-induced ageing rats. However, DPSC treatment significantly reduced collagen accumulation compared to that in the D-gal group (Figs. 4C and 4D).

Table 2 DPSC transplantation improved the echocardiogram (ECG) changes in D-gal-induced ageing rats.

	Control	D-gal-treated	D-gal + DPSC-treated	
P-R interval (s)	.0432 ± .0002	.04387 ± .0055	.0478 ± .008	
P duration (s)	.0240 ± .001	.01159 ± .0126	.02616 ± .01264	
cQT interval (s)	.4567 ± .017	.705 ± .02a	.504 ± .027b	
QRS duration (s)	.01125 ± .001	.0150 ± .0001a	.0122 ± .0007b	
R amplitude (mv)	.2813 ± .0136	.1041 ± .048a	.24270 ± .047b	
ST Height (mv)	.118 ± .002	.0298 ± .0028a	.0601 ± .009b	
Notes.

a p < 0.05, compared with the control group.

b p < 0.05, compared with the D-gal group.

Data are expressed as mean ± SEM. N = 10 per group.

DPSCs migrate into and survive in the cardiac tissue with a few transplanted cells differentiated into cardiomyocytes

An immunofluorescence-based examination of the cardiac tissue of D-gal + DPSCs rats was performed to assess the engraftment of injected DPSCs into the heart. Transplanted DPSCs were distinguished from recipient cells via labelling with PKH26, which is a cell membrane-binding dye with red fluorescence. PKH26-labelled cells were distributed in the cardiac tissues of the D-gal + DPSCs rats (Figs. 5B, 5F, 5J, and 5N). Interestingly, a few of the transplanted cells differentiated into cardiomyocytes, as indicated by the colocalization of cardiac troponin T (cTnT) (Figs. 5A–5H). or cardiac troponin I (cTnI) with PKH26-labelled cells (Figs. 5I–5P).

Figure 3 DPSCs upregulated connexin 43 expression in D-gal-induced ageing heart at 2 weeks post-transplantation of the last DPSC injection.

Representative connexin-43 immunofluorescence images (A–C) and number of connexin-43 spots in ventricular cardiac tissue (D) of control, ageing (D-gal), and transplanted (D-gal + DPSCs) rats. ***P < 0.001 compared with the control group; ###P < 0.01 compared with the D-gal group. Data are expressed as mean ± SEMs. N = 10 per group. Scale bar = 500 µm.

Figure 4 Transplanted DPSCs improved the cardiac histopathological alterations and cardiac fibrosis in D-gal-induced ageing rats at 2 weeks post-transplantation of the last DPSC injection.

Representative haematoxylin and eosin staining of cross (A) and longitudinal (B) section of the left ventricle. Representative Masson’s trichrome staining and extent of fibrosis (C and D) in the control, ageing (D-gal) and transplanted (D-gal + DPSCs) groups. **P < 0.01, ***P < 0.001, compared with the control group; #P < 0.05, ##P < 0.01, compared with the D-gal group. Data are expressed as mean ± SEM. N = 10 per group. Scale bars = 100 µm (A) and 500 µm (C).

Figure 5 Survival and differentiation of transplanted DPSCs into cardiac cells in the transplanted (D-gal + DPSCs) group at 2 weeks post-transplantation of the last DPSC injection.

Some PKH26-labelled DPSCs coexpressed cardiac troponin T (cTNT) (A–D) or cardiac troponin I (cTNI) (I–L). The boxed areas in A–D and I–L appear at a higher magnification in E–H and M–P, respectively. Inserts show a higher magnification of the boxed regions in E–H and M–P. PKH26-labelled cells (red) B, F, J, N; cTNT-positive cells (green) (C, G) cTNI-positive cells (green) K, O; DAPI-stained nuclei (blue) A, E, I, M. Merged images (D, H, L, P). Scale bars = 100 µm (A–D and I–L), and 50 µm (E–H and M–P).

DPSCs upregulate Sirt1 expression and exert an antioxidative effect in D-galactose-induced cardiac ageing in rats

We further investigated whether DPSC transplantation has antioxidant effects. Compared with the control rats, we observed that aged rats exhibited higher concentrations of MDA (Fig. 6A), whereas the SOD (Fig. 6B) and GSH (Fig. 6C) levels were dramatically reduced in aged hearts compared with those in control rats. In D-gal + DPSC-treated rats, MDA concentrations in the cardiac tissue were significantly reduced (Fig. 6A), whereas SOD (Fig. 6B) and GSH (Fig. 6C). levels were higher than those in the aged rats (Figs. 6A–6C). Sirt1 demonstrated its ability to delay ageing and induce cardiac antioxidative effects (Alcendor et al., 2007; Cencioni et al., 2015; Lu et al., 2014; Ministrini et al., 2021). Interestingly, DPSC-treated rats exhibited higher Sirt1 expression than D-gal-treated rats (Fig. 7A).

Figure 6 DPSCs induced antioxidative effect on D-gal-induced ageing rats at 2 weeks post- transplantation of the last DPSC injection.

Status of malondialdehyde (MDA, A), superoxide dismutase (SOD, B), and glutathione (GSH, C) in the hearts of control, aged (D-gal), and transplanted (D-gal + DPSCs) rats. *P < 0.05, ***P < 0.001, compared with control; ###P < 0.001 compared with the D-gal group. Data are expressed as mean ± SEM. N = 10 per group.

Figure 7 DPSCs upregulated Sirt1 expression and exerted anti-apoptotic effects in D-gal- induced aged hearts at 2 weeks post-transplantation of the last DPSC injection.

(A) Western blotting analysis showing the expressions of Sirt1 and apoptotic markers in the hearts of control, aged (D-gal), and transplanted (D-gal + DPSCs) rats. (B) Densitometry analysis of apoptosis- associated protein levels in different experimental groups. *P < 0.05, **P < 0.01, ***P < 0.001 compared with control; ###P < 0.001 compared with the D-gal group. Data are expressed as mean ± SEM. N = 10 per group.

DPSCs exhibit anti-apoptotic effects in D-galactose-induced cardiac ageing in rats

Cardiac ageing is linked to a marked reduction in Bcl2, an anti-apoptotic marker, and a significant elevation in Bax and cytochrome c, thereby triggering apoptosis (Pollack et al., 2002) Western blot analysis demonstrated that the mitochondria triggered an increase in the pro-apoptotic markers Bax (Fig. 7B), cytochrome c (Fig. 7D), and cleaved caspase-3 (Fig. 7E), whereas the anti-apoptotic marker Bcl-2 (Fig. 7C) was reduced in D-gal-treated rats. However, in D-gal + DPSC-treated rats, the expression of all upregulated apoptotic markers decreased (Figs. 7B, 7D, and 7E), while the expression of Bcl-2 was enhanced (Fig. 7C).

Effects of DPSC administration on senescence-associated markers in D-galactose-induced ageing rats

Senescence-linked β-galactosidase (SA-β-gal) activity is commonly used to recognize cells as senescent. A group of cell cycle regulatory factors, including P21, and P53, are also considered as senescence markers (Shimizu & Minamino, 2019). The expression of senescence-related markers, such as SA- β-gal (Fig. 8A), p53 (Fig. 8B) and p21 (Fig. 8C), was significantly upregulated in aged rats compared to those in control rats. In D-gal + DPSC-treated rats, DPSCs efficiently reduced the expression of all evaluated ageing markers (Figs. 8A–8C).

Figure 8 Protective effects of DPSCs assayed on senescence-associated markers in D-gal-induced aged heart at 2 weeks post-transplantation of the last DPSC injection.

Gene expression of senescence-associated markers such as senescence-linked β-galactosidase (SA-β-gal, A), p53 (B), and p21 (C) in the heart of control, aged (D-gal), and transplanted (D-gal + DPSCs) rats as measured by RT-qPCR. ***P < 0.001 compared with control; ###P < 0.001 compared with the D-gal group. Data are expressed as mean ± SEM. N = 10 per group.

Discussion

Ageing is characterized by the progressive aggregation of cellular damage, resulting in the gradual dysfunction of several organs. Age-associated alterations in the heart include left ventricular hypertrophy, upregulated collagen deposition, and cardiac dysfunction. Ageing contributes significantly to the progression of age-associated cardiovascular diseases, leading to a notably elevated prevalence of such conditions in elderly populations. Therefore, there is an immediate need to develop more effective therapeutic approaches to manage cardiac insufficiency associated with ageing (Yan et al., 2021; Xie et al., 2023). A substantial body of evidence suggests the potential of MSCs in the treatment of heart disease (Jeong et al., 2018; Fu & Chen, 2020; Meng et al., 2022; Kalou, Al-Khani & Haider, 2023). DPSCs exhibit a range of distinct biological characteristics that confer protective effects on injured tissues, including the cardiac tissue (Gandia et al., 2008; Sui et al., 2020; Song et al., 2023). The current study aimed to investigate, for the first time, the potentially favourable outcomes of the systemic injection of DPSCs in D-galactose (D-gal)-induced aged hearts.

Histopathological examination of ageing cardiac tissue revealed cardiomyocyte hypertrophy and enhanced LV fibrosis, consequently reducing LV elasticity and leading to cardiac dysfunction. Ageing hearts exhibit distinctive histological and functional characteristics, such as increased cardiac remodelling and declining cardiac reserve. Comparable with normal ageing in rats, D-gal-induced ageing models present cardiac structural and functional ageing alterations (Lazzeroni et al., 2022; Wang et al., 2022). In the current study, we demonstrated that D-gal-induced ageing rats had aberrant cardiac structure and enhanced collagen deposition in the perivascular and interstitial areas of the heart. These histopathological alterations in the cardiac structures were reflected in our echocardiography analysis, which revealed an increase in LVIDd and EDV; however, no significant difference was observed when compared with control rats. In contrast, LVIDs, ESV, and LVPWd were markedly increased in aged rats. Echocardiography also revealed worsening cardiac function, as indicated by decreased FS% and ES%. Interestingly, the observed morphological and functional alterations were dramatically improved in the DPSC transplanted group. Our results are in line with those of previous studies, albeit using other source types of MSCs. It has been documented that the systemic injection of adipose-derived stem cells (ADSCs) significantly ameliorated structural alterations and cardiac performance in D-gal-induced ageing rats (Chang et al., 2021). Significant histopathological improvement was also observed in D-gal-induced ageing rats administered intravenous injections of Wharton’s jelly stem cells, as indicated by a reduction in collagen and reversal of ageing-induced structural changes (Hu et al., 2022). Intracardiac injection of BMMSCs was also found to ameliorate natural ageing-associated cardiac hypertrophy and fibrosis and enhance cardiac performance, as reflected by increased EF%, FS%, and reduced LVIDs (Zhang et al., 2015).

In this study, the cardiac dysfunction observed in aged rats was combined with significant changes in the ECG patterns. Aged rats exhibited an increased QRS duration and cQT interval, along with a decreased R amplitude and ST height compared to that observed in the control group. A decline in the expression of the vital gap junction protein connexin-43, which is responsible for transmitting signals along the conduction pathway and forming a functional syncytium between myocytes (Rodriguez-Sinovas et al., 2021), was also observed. The impact of disturbed functional syncytium on conducting signals could be exacerbated by the loss of cardiomyocytes, which naturally develops during ageing, along with accompanying reactive fibrosis enhancement (Tracy, Rowe & LeBlanc, 2020). In our study, DPSC transplantation reduced fibrosis in aged rats and preserved connexin-43 expression, thereby improving ECG alterations. The therapeutic effects of MSCs on cardiac electrical conductivity have also been documented. Li et al. (2018) observed that BMMSC transplantation increased the density of connexin 43, which improved the dispersion of electrical excitation in rats with myocardial infarction. MSC therapy also resulted in a shorter QRS duration and QTc interval, indicating the capability of MSCs to improve cardiac electrical velocity in a murine double infarction model (Park et al., 2022).

Most cardiomyocytes do not undergo active proliferation, and the yield rate of cardiomyocytes in the human heart is less than or equal to 1% per year (Gude et al., 2018). Although cardiac progenitor cells (CPCs) are considered an interesting source of cell therapy, the aged human heart exhibits a remarkable population of senescent CPCs, which may be responsible for the development of cardiac malfunction instead of promoting regenerative effects (Shimizu & Minamino, 2019). Previous research has shown that MSCs from various sources possess the ability to differentiate into cardiomyocytes (Toma et al., 2002; Guo et al., 2018; Soltani & Mahdavi, 2022; Zhou et al., 2023). Whether MSCs can differentiate into cardiomyocytes remains debatable. While some studies have speculated that the detected regeneration could be a result of donor cell fusion with recipient cardiomyocytes (Alvarez-Dolado et al., 2003), other studies have reported that BMMSCs can give rise to cardiomyocytes under laboratory conditions, even during a lack of cardiomyocytes, thus excluding the likelihood of cell fusion (Pallante et al., 2007). Notably, MSCs derived from various sources are considered safer than genetically engineered MSCs due to their reduced risk of oncogenicity or malformation (Hatzistergos et al., 2011). MSCs of different origins exhibit varying capacities for differentiation into cardiomyocytes. The specific differentiation potential of MSCs from specific sources may lead to more favourable outcomes compared to others (Park et al., 2022). To our knowledge, no previous study has examined the differentiation capacity of DPSCs in a cardiac ageing model. In our study, a considerable number of DPSCs migrated and survived in the cardiac tissue, and some cells colocalized with cTnT and cTnI. Our results are in line with those of several studies reporting the capability of DPSCs to differentiate into cardiomyocytes (Arminan et al., 2009; Xin et al., 2013; Potdar & Jethmalani, 2015; Sung et al., 2016; Al Madhoun et al., 2021). However, our results suggested that only a small number of DPSCs differentiated into cardiomyocytes. To further investigate the underlying mechanisms by which the transplanted cells improved the structural and functional alterations in D-gal-induced ageing hearts, we sought to evaluate the potential paracrine effects of the injected DPSCs.

Emerging evidence consistently supports the notion that oxidative stress plays a significant role in the physiological progression of ageing (Maldonado et al., 2023). When exposed to elevated levels of oxidative stress, p53 displays pro-oxidative activities, intensifying stress levels, and ultimately triggering apoptosis (Liu & Xu, 2011; Shi et al., 2021). A critical determinant in the ageing process of cardiac tissue is the equilibrium between anti-apoptotic Bcl-2 and pro-apoptotic Bax proteins. Bcl-2 inhibits apoptosis by suppressing cytochrome c release from mitochondria (Pollack et al., 2002). Sirtuins, which are a group of nicotinamide adenine dinucleotide (NADþ)-dependent histone deacetylases, are thought to regulate cardiac ageing by influencing mitochondrial stress responses. Sirt1 activation has been suggested to increase the expression of antioxidant enzymes, such as SOD and catalase, leading to the inhibition of ROS generation, which is a key player in oxidative stress (Alcendor et al., 2007; Cencioni et al., 2015). In our study, ageing hearts demonstrated increased oxidative stress and increased expression of p53, pro-apoptotic BAX, and cytochrome c, and a decrease in the antioxidant markers SOD, Bcl-2, and Sirt1. DPSC transplantation significantly ameliorated these changes. Our results are in line with those of previous studies. Various MSC sources have demonstrated the ability to exhibit antioxidant capacity, primarily through the upregulation of glutathione transferase (GST) and the increased activity of SOD. This, in turn, modulates the genes driven by the antioxidant response element (Xia et al., 2021). In studies involving D-gal-stressed H9c2 cells, the culturing of these cells with a conditioned medium derived from MSCs led to significant improvements in cell viability, upregulation of SOD expression, downregulation of oxidative stress levels, and inhibition of p53 activity (Hu et al., 2022). In the context of D-gal-induced ageing in rats, adipose tissue-derived mesenchymal stem cells (ADMSCs) exhibited a downregulation in mitochondria-triggered apoptotic markers, including Bax, cytochrome c, and cleaved caspase-3. Conversely, it has been reported that ADMSCs upregulate Bcl-2 and Sirt1 expression compared to other treatment groups (Chang et al., 2021). Our findings support those of previous studies demonstrating the antioxidant and anti-apoptotic capabilities of DPSCs (Song, Jue SS. Cho & Kim, 2015; Ullah et al., 2018; Hernandez-Monjaraz et al., 2020; El-Kersh, El-Akabawy & Al-Serwi, 2020; Al-Serwi, El-Kersh & El-Akabawy, 2021). Our results demonstrated that antioxidant and anti-apoptotic capabilities of transplanted DPSCs are mediated by downregulation of the expression of p53, pro-apoptotic BAX, and cytochrome c and upregulation of the antioxidant markers SOD, Bcl-2, and Sirt1. In cultured cardiac myocytes, conditioned medium derived from DPSCs significantly inhibits apoptosis under hypoxic/serum deprived conditions. It also reduced the expression of proinflammatory genes induced by lipopolysaccharide. Notably, the anti-apoptotic effect of the conditioned medium of DPSCs was found to be more efficacious than that of conditioned media derived from BMSCs or ADSCs on cardiac myocytes (Yamaguchi et al., 2015).

Conclusions

Our results reveal that the systemic injection of DPSCs ameliorated cardiac structural and functional alterations occurring in a rat model of D-gal-induced cardiac ageing. Our findings support the beneficial effects of intravenously transplanted DPSCs on ageing-related cardiac structural and functional changes. However, to examine whether the transplanted cells could reverse ageing-induced changes, further studies are needed wherein DPSCs should be transplanted after cardiac ageing has been established. In addition, before initiating clinical trials, additional studies are required to promote the differentiation of these cells into cardiomyocytes in vivo. Furthermore, a deeper understanding of the underlying mechanisms responsible for the favourable outcomes of DPSC-secreted factors is crucial.

Supplemental Information

Figure S1 Characterization of the dental pulp stem cell (DPSC) population

Supplemental Information 2 Raw Data for Figures 2, 3 and 4

Supplemental Information 3 Raw Data for Figures 6, 7 and 8

Supplemental Information 4 Raw data of transthoracic echocardiography of different experimental groups

Supplemental Information 5 H & E images of different experimental groups

Supplemental Information 6 Masson’s staining images of different experimental groups

Supplemental Information 7 Full-Length uncropped images

Supplemental Information 8 Arrive Checklist

Additional Information and Declarations

Competing Interests

Author Contributions

Animal Ethics

Data Availability

The authors declare there are no competing interests.

Gehan El-Akabawy conceived and designed the experiments, performed the experiments, analyzed the data, prepared figures and/or tables, authored or reviewed drafts of the article, and approved the final draft.

Sherif Othman Fathy El-Kersh conceived and designed the experiments, performed the experiments, analyzed the data, prepared figures and/or tables, authored or reviewed drafts of the article, and approved the final draft.

Ahmed Othman Fathy Othman El-Kersh conceived and designed the experiments, performed the experiments, analyzed the data, prepared figures and/or tables, authored or reviewed drafts of the article, and approved the final draft.

Shaimaa Nasr Amin performed the experiments, analyzed the data, prepared figures and/or tables, authored or reviewed drafts of the article, and approved the final draft.

Laila Ahmed Rashed performed the experiments, analyzed the data, prepared figures and/or tables, and approved the final draft.

Noha Abdel Latif performed the experiments, analyzed the data, prepared figures and/or tables, authored or reviewed drafts of the article, and approved the final draft.

Ahmed Elshamey performed the experiments, analyzed the data, prepared figures and/or tables, and approved the final draft.

Mohamed Abdallah Abd El Megied Abdallah performed the experiments, analyzed the data, prepared figures and/or tables, and approved the final draft.

Ibrahim G Saleh analyzed the data, prepared figures and/or tables, and approved the final draft.

Zaw Myo Hein analyzed the data, prepared figures and/or tables, and approved the final draft.

Ibrahim El-Serafi analyzed the data, prepared figures and/or tables, and approved the final draft.

Nabil Eid analyzed the data, prepared figures and/or tables, and approved the final draft.

The following information was supplied relating to ethical approvals (i.e., approving body and any reference numbers):

All experimental procedures involving animals were approved by the Institutional Review Board of Ajman University, UAE [IRB# M-F-A-11-Oct].

The following information was supplied regarding data availability:

The raw measurements are available in the Supplementary Files.

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
