# Peer review of "Dental pulp stem cells ameliorate D-galactose-induced cardiac ageing in rats"

_PeerJ, doi:10.7717/peerj.17299_

## Round 0.1 · original submission · Major Revisions

Both reviewers have raised important issues regarding the data and conclusion as described in the manuscript. Hence it is recommended to carefully address each and every comment by reviewer 1 and 2. In particular, the results section and corresponding figures and legends require more detailed information and some corrections.

As additional comments, there is a mistake in the title to Figure 8 which states “Protective effects of DPSCs on survival markers in D-gal-induced aged heart” whereas the markers analysed are not “survival markers” but instead, as correctly described in the results section, they are “senescence-associated markers” and are shown indeed as increased in cardiac tissue from D-gal-induced aging. Another correction needed is in the legend to Figure 5 stating “Number of PKH26-labelled DPSCs co-expressing cardiac troponin T and cardiac troponin I” : either some numbering of labelled DPSCs co-expressing Troponin T or I should be provided (which is not the case) or the beginning of this title: “Number of” must be removed. It would be important to also specify in each figure legend the timing of the last DPSC injection before analysis.

The conclusion paragraph needs to be tuned down in the sense that was underlined in Reviewer’s 2 comment: the data presented do not actually support that DPSC would be efficient as a preventive therapy for age-associated cardiovascular diseases. The authors should indicate what further work is required to support such a possibility.

Reviewer 1 ·

Basic reporting

The manuscript under review presents a robust and well-crafted document with an innovative perspective on advancing novel treatments to combat aging cardiac disease, by implanting Dental pulp stem cells in ameliorate D-galactose induced cardiac ageing in rats. The writing is succinct and clear, with minor grammar revision, featuring a well-supported literature introduction and solid data.
1. Grammer- well written text, minor revisions were suggested in the annotated manuscript.
2. Introduction will benefit from expanding on marker biology (line 79), why are the markers important to your research question?
3. Line 114, explain statement. Why are these processes being important? Are they hallmarks of health young DPSCs?
4. Line 118, note in short on the D-gal aging cardiac model.

Experimental design

nicely outlined primary research within Aims and Scope of the journal, the Research question is well defined (line 123), data presented answered the authors specific hypothesis and had significant predictive benefit which is both relevant & meaningful. Methods were described well and were relevant to the data presented, please see comments in the manuscript text.
5. Consider moving Figure 1 to supplementary section.
6. Results- Refer to specific sub figure section when discussing results, please see annotated text for reference.
7. Figure 3- add titles for each tissue. what is the control? add arrows to point to the abnormality listed.
8. Section 3.5 (line 348-354)- add short explanation to what PKH26 function.
9. Figure 5- add to the figure images titles of what tissue was stained, Control/D-gal/D-gal+DPSC...
10. Figure 6/ Figure7/ Figure 8- add subsection to the figures.
11. Discussion, line 488, Could you hypothesis how DPSCs could facilitate anti-apoptotic events? what mechanism?

Validity of the findings

Conclusions are well stated, linked to original research question and well supported with literature.

Annotated reviews are not available for download in order to protect the identity of reviewers who chose to remain anonymous.

·

Basic reporting

The study was carried out with the objective of examining the structural and functional changes associated with D galactose induced aging and the cardiac response to dental pulp stem cell (DPSC) transplantation. The experiments were presumably carried out in 8 week old Sprague-Dawley (SD) rats. D-galactose (300)mg/kg/day) was administered intraperitonially for 8 weeks and DPSCs (1X 106 was intravenously injected every 2 weeks. Cardiac structural changes were assessed using histological techniques and immunohistochemistry. Functional changes were evaluated using echocardiography and ECG. Transplanted cells homed in to the cardiac tissue and differentiated into cardiomyocytes. Improvement in cardiac morphology and function was observed on transplantation of DPSCs to D-gal treated rats. Upregulation of antioxidative parameters and attenuation of senescence and apoptotic effects were seen
A significant observation in the study is the differentiation of transplanted DPSCs to cardiomyocytes and the beneficial effects of these cells in the mitigation of cardiac injury in experimental aging. The study is important as cell therapy for cardiac repair is gaining momentum. The experimental design and animal model are appropriate to show the beneficial effects of DPSCs in heart. However, transplantation of stem cells to prevent aging induced cardiac damage is not a viable suggestion.

Experimental design

The sample size and animal species used for the study are appropriate. Details of the study and evaluation of variables are provided in sufficient detail.
The points that need consideration are:
1. It is mentioned that 8 week old rats were purchased. Generally experimental animals are acclimatized to laboratory conditions before initiation of experiments. Hence the age at the start of the experiment has to be included.
2. Line 135- It is mentioned that rats should be euthanized in the event of rapid weight loss. Therefore a weight chart would have been maintained. It will be good to include the body weight at initiation and on completion of the study.
3. The intravenous site chosen for administration of DPSCs has to be mentioned.
4. Left ventricular hypertrophy is a feature of the aging heart (Introduction- Line 70.) increase in heart weight is a well-known indicator of ventricular hypertrophy. Hence heart weight is an essential variable, and has to be included either as a fraction of body weight or preferably as ratio of tibial length.
5. Line 261- The zone selected for histopathological studies have to be mentioned

Validity of the findings

Results
The general presentation is good. But, lacunae have been noticed.
1. The statistical comparison of control with the D- gal + DPSCs group has to be made for all variables
2. As given above, inclusion of heart weight or left ventricular weight is essential.
3. Line 306 and Fig.1- The passage of which the cells were characterized for stem cell markers should be mentioned. Ideally the cells of the passage used for transplantation should be examined.
4. Reference to other studies should be included in the ‘Introduction’ (Lines 326 & 363-364)
5. Fig. 4 – The scale bars do not comply with the magnification. The difference between the 2 rows of ‘A’ are not mentioned. The name of the group has to be given in the representative pictures

Discussion and Conclusion-
The claim that systemic injection of DPSCs ameliorates cardiac structural and functional alterations in a rat model of D-gal-induced cardiac aging is correct. The potential therapeutic use of intravenously transplanted DPSCs in the prevention of aging-related cardiac structural and functional alterations is not practical; as transplantation of stem cells in a healthy individual in anticipation of age associated cardiac dysfunction cannot be supported. Ideally experiments should show reversal of morbidity in dysfunctional heart. In view of the positive changes observed in response seen following transplantation of DPSCs amelioration of cardiac dysfunction of aging heart can be anticipated.

Minor points-
1. Fig. 2- The labels of the representative pictures should be mentioned alongside the variables in the legend Eg. B. Ejection fraction
2. Line 113- The sentence needs correction.
3. Line 328- Use of the term ‘restored’ would be better than ‘rescued’
4. Line 344- Interstitial and perivascular areas of ‘myocardium’ and not cardiomyocytes.
5. Line 404-405 The sentence needs correction
6. Line 407- Enhanced collagen deposition may be better than predisposition

Additional comments

Abstract- Objectives of the study along with brief description of the methodology and prominent finding are presented.
The species of rat used for the study and age have to be included.

Introduction- The background of the study is stated clearly and the literature is presented in an appropriate fashion.

The last sentence (Lines 123-125) mentions that the study was carried out to examine the possible efficacy of an intravenous injection of DPSCs in a D-gal-induced rat model of cardiac ageing to evaluate their potential as a preventive therapy for age-associated cardiovascular diseases. There are concerns in the statement.
1. Aging being a normal phenomenon, preventive therapy implies that DPSCs have to be transplanted to all individuals above a certain age. This is not practical.
2. The study helps to assess the beneficial effects of DPSC transplantation on cardiac tissue. To examine whether the transplanted cell reverse aging induced changes, DPSCs have to be transplanted only after cardiac aging has been established.

---

## Round 0.2 · Minor Revisions

The manuscript will be acceptable for publication provided few further minor corrections are made as requested by reviewer 2 and as I indicated in the attached PDF document entitled "peerj-93290-Track_changed_Revised_El-Akabawy-manuscript-DPSCs-Cardiac-aging_edits-AF". These corrections include: a grammar correction in the abstract (line 54); two important corrections in the results section 3.2 : in the subtitle line 412 replace BMMSC by DPSC; in line 417 replace "improved" by "reduced"and in the subtitle 3.3 line 419: replace "improves" by "reduces". In the results section 3.6, line 495-496 : reword the subtitle as indicated and make the correction line 499 (English grammar) and last, in the figure legends, title to figure 7, line 1166 "Protective effects of DPSCs on senescence-associated markers" must be changed to "Protective effects of DPSCs assayed on senescence-associated markers"

Reviewer 1 ·

Basic reporting

The authors' skillful integration of the suggested corrections has substantially improved the clarity and understanding of both the manuscripts and the accompanying data. These revisions not only enhance the overall quality of the material but also offer profound insights into the potential of dental pulp stem cells in mitigating D-galactose-induced cardiac aging in rats. Such insights carry considerable significance for researchers and clinicians, enriching our comprehension and potentially paving the way for novel diagnostic and therapeutic approaches in the realm of cardiac aging

Experimental design

all comments were addressed and implemented in the revised manuscript

Validity of the findings

all comments were addressed and implemented in the revised manuscript

Additional comments

all comments were addressed and implemented in the revised manuscript

·

Basic reporting

The English language can be improved

Experimental design

The research question is well defined and experimental design is appropriate

Validity of the findings

Required corrections are made in the results and the conclusion is also revised appropriately

Additional comments

The Authors have addressed the queries raised by Editor and the Reviewers
The following minor errors are to be corrected
Line 170- ‘Cells were used at passage 4’; to be corrected as, ‘Cells of Passage 4 were evaluated’
Line 344-345 - ‘transplanted cells improved D-gal-induced hypertrophy’ can be corrected as, ‘transplanted cells attenuated D-gal-induced hypertrophy.’
Lines 378 and 383 - Figure numbers have to be included.
Lines 424-425 Use of term ‘such as’ is not appropriate
Line 485 – “Comparable with normal aging” is more appropriate than “Compared with normal ageing”
Figures 4, 6 and 7 Significance indicators (* and #) are not defined correctly

---

## Round 0.3 · accepted · Accept

All of the reviewers' comments and suggestions have now been addressed thus making this manuscript acceptable for publication.